# Should Neurogenic Supine Hypertension Be Treated? Insights from Hypertension-Mediated Organ Damage Studies—A Narrative Review

**DOI:** 10.3390/biomedicines14010040

**Published:** 2025-12-24

**Authors:** Cristiano Fava, Federica Stocchetti, Sara Bonafini

**Affiliations:** Section of General Medicine C, Department of Medicine, University and Azienda Ospedaliera Universitaria Integrata of Verona, 37126 Verona, Italy

**Keywords:** neurogenic supine hypertension, neurogenic orthostatic hypotension, synucleinopathies, Parkinson’s disease, antihypertensive therapy, hypertension-mediated organ damage

## Abstract

Neurodegenerative synucleinopathies—including Parkinson’s disease, multiple system atrophy, pure autonomic failure, and dementia with Lewy bodies—often feature cardiovascular autonomic dysfunction. Neurogenic orthostatic hypotension (nOH) is common and symptomatic, while neurogenic supine hypertension (nSH) is less frequent but may carry long-term cardiovascular risks. Lifestyle measures are first-line for managing nSH, yet persistent hypertension unresponsive to nonpharmacological strategies presents a treatment dilemma. Limited trial data and unclear guidelines make it difficult to determine when antihypertensive therapy is appropriate. Evidence from studies on hypertension-mediated organ damage (HMOD)—assessed through markers such as carotid intima-media thickness, pulse wave velocity, left ventricular hypertrophy, estimated glomerular filtration rate, and white matter hyperintensities—suggests that nSH, rather than the underlying neurodegenerative disorder, drives vascular, cardiac, renal, and cerebral injury. Therefore, treatment decisions should be individualized. While antihypertensive therapy may help prevent subclinical organ damage, clinicians must balance this benefit against the risk of worsening nOH and further compromising overall prognosis.

## 1. Introduction

Neurodegenerative disorders such as Parkinson’s disease (PD), multiple system atrophy (MSA), pure autonomic failure (PAF), and dementia with Lewy bodies (DLB) are collectively called synucleinopathies since are characterized by the abnormal accumulation of alpha-synuclein protein into clumps called Lewy bodies (in neurons) or glial cytoplasmic inclusions (in glia). Affected patients often present with cardiovascular autonomic dysfunction [1]. This commonly leads to neurogenic orthostatic hypotension (nOH), which may occur alone or in combination with neurogenic supine hypertension (nSH). As a result, these patients may experience two opposing blood pressure-related issues that not only reduce quality of life but can also cause serious health complications—ranging from falls due to hypotension to organ damage or cardiovascular events due to hypertension.

General practitioners and specialists alike may encounter such patients and face a complex clinical dilemma for which there is no universally accepted solution. Although several position and consensus papers provide guidance on management, they are largely based on expert opinion, as robust observational studies and randomized clinical trials on cardiovascular outcomes in this field are largely lacking [2,3].

In this context, intermediate phenotypes—such as atherosclerotic or arteriosclerotic markers—may offer valuable insights into whether nSH poses a significant cardiovascular risk, similar to that seen in patients with essential hypertension. In recent years, particular attention has been paid to nocturnal (and therefore often supine) hypertension, which is increasingly recognized as one of the most harmful forms of arterial hypertension [4,5].

In this review, we have compiled the limited available studies that focus on established, non-invasive subclinical cardiovascular indices. Our aim is to support clinicians in managing the complex care of patients affected by both nOH and nSH.

## 2. Definition of Supine Hypertension in the Context of Autonomic Failure

According to the consensus statement of the American Autonomic Society and the European Federation of Autonomic Society, in patients with neurogenic nOH, SH is defined as a systolic blood pressure (BP) and/or a diastolic BP ≥ 140/90 mmHg, respectively, measured after at least 5 min of lying in the supine position.

A different entity is the nocturnal hypertension, which is detected only with ambulatory BP monitoring (ABPM) and is defined as nocturnal systolic BP ≥ 120 and/or diastolic BP ≥ 70 mmHg, according to European guidelines [5].

Related, although distinct, is the concept of reverse dipping, which is the pattern characterized by the inversion of the normal day-to-night BP ratio, with nighttime BP values higher than daytime ones [6].

Supine hypertension may occur both during the day and during the night and depends on the lying position, whereas nighttime hypertension is both a direct consequence of recumbent position in patients with nSH, but could also be provoked by sleep disturbance, obstructive sleep apnoea, obesity, high salt intake in salt-sensitive subjects, chronic kidney disease (CKD), diabetic neuropathy, and old age [4].

A non-dipping pattern often coexists in patients with SH, however a normal dipping pattern has also been reported in about one-third of patients suffering from supine hypertension [7].

## 3. Epidemiology of Autonomic Failure Disorders

Among neurodegenerative disorders associated with cardiovascular autonomic failure, Parkinson’s disease (PD) is the most prevalent, affecting over 8.5 million people worldwide (GBD 33069326).

Mortality is high in patients with PD [8], with an estimated 5% reduction in survival for each year of follow-up. According to the World Health Organization (WHO), PD accounted for 5.8 million disability-adjusted life years (DALYs) and over 300,000 deaths globally in 2019—figures that have steadily increased compared to previous estimates (WHO Global Health Estimates (https://www.who.int/data/gho/data/themes/mortality-and-global-health-estimates/global-health-estimates-leading-causes-of-dalys) accessed 26 May 2025) [9]. Mortality in PD is primarily linked to neurological complications, with cardiovascular-specific mortality being relatively uncommon.

A large, population-based prospective cohort study in Korea compared 8220 PD patients with 41,100 age- and sex-matched controls without PD. PD patients had more than twice the risk of death compared to non-PD individuals. Among these deaths, only 15% were attributed to circulatory system complications, while the majority were due to neurological causes [10]. Smaller studies have confirmed that neurological complications remain the leading cause of death in PD [11].

DLB is an increasingly recognized cause of dementia in the elderly, with an estimated community prevalence of up to 4% [12]. DLB is occasionally associated with autonomic failure, particularly nOH [13].

Less common neurodegenerative conditions involving cardiovascular autonomic failure include multiple system atrophy (MSA) and pure autonomic failure (PAF). MSA, another alpha-synucleinopathy, is characterized pathologically by alpha-synuclein aggregates in oligodendrocyte cytoplasm [14]. Clinically, MSA presents with varying degrees of autonomic dysfunction, parkinsonism, ataxia, and corticospinal tract degeneration [14]. Its prevalence is approximately ten times lower than PD, though underdiagnosis has been suggested [15,16]. Prognosis is poor, with over two-thirds of MSA patients dying within five years of diagnosis.

PAF is characterized by the initial presentation of autonomic symptoms, without overt motor or cognitive involvement. Over time, some patients with PAF may convert to PD, DLB, or MSA, while others remain stable [17]. Patients with persistent PAF generally have a more favorable prognosis and longer survival, especially if their disease does not progress to another alpha-synucleinopathy.

## 4. Parkinson’s Disease or Other Alpha-Synucleinopathies and Hypertension Incidence/Prevalence

While one of the primary concerns in PD is nOH—which significantly impairs quality of life by increasing the risk of falls, syncope, and functional disability—high blood pressure is also commonly observed in these patients.

Several studies conducted in the USA and the UK have reported a lower prevalence of hypertension among PD patients with respect to matched controls and someone suggested even a protective role for hypertension or antihypertensive medications [18,19,20]. However, a recent Chinese case–control study, involving 671 PD patients and 671 age- and sex-matched controls (mean age: 64 years), reported a higher prevalence of hypertension, reaching 46% in PD [21]. Although these studies have notable limitations—such as reliance on self-reported questionnaires and medical records for diagnosing hypertension—they still highlight the fact that a substantial proportion of PD patients are hypertensive, raising important questions about whether and how to treat hypertension in this population, especially considering the complex circadian variation in blood pressure patterns in PD.

Another study focusing on early-stage PD reported an even higher prevalence of nSH, which was significantly associated with nOH, advancing age, and pre-existing hypertension [22]. In more advanced stages of PD or MSA, the prevalence of SH increases further, reaching up to 34% [2]. The proportion is even higher among patients with pure autonomic failure (PAF) [23,24].

A recent systematic review and meta-analysis evaluating hypertension patterns through ambulatory blood pressure monitoring (ABPM) found that approximately 38% of PD patients had daytime hypertension and a similar proportion had clinostatic or nocturnal hypertension. Moreover, nSH was present in 27% of cases and abnormal circadian dipping patterns were also common, with 40% classified as reverse dippers and 36% as reduced dippers [25].

Another meta-analysis of longitudinal studies (7 studies, including 3170 individuals who developed PD and 339,517 controls) investigated whether hypertension prior to PD onset might be a risk factor for the disease. The results indicated that preexisting hypertension was associated with an 80% increased risk of developing motor-stage PD [26].

An interesting longitudinal study evaluated 267 early Parkinson’s disease (PD) patients twice over a three-year interval using head-up tilt testing identifying three distinct groups based on their positional blood pressure (BP) profiles: (i) BP-extreme that is patients exhibiting either nSH or nOH; (ii) BP-mild that is patients with milder dysregulation, such as delayed nOH or orthostatic hypertension, while remaining normotensive; and (iii) BP-none that is patients without any identifiable BP disturbances.

Over a mean follow-up period of 29 months, the study reported a significant increase in the BP-extreme phenotype, rising from 79 to 121 patients. The most notable increase was in the prevalence of nOH, which rose from 18% to 32.5%, while the prevalence of nSH slightly declined (from 8.6% to 6.8%) [27].

About the comparison of different alpha-synucleinopathies, a recent systematic review and meta-analysis including 19 studies (nearly 5000 patients in total) examined the prevalence of nSH [28]. The total prevalence was 30.4%, with the highest rates observed in DLB and PAF (nearly 50%) and somewhat lower rates in MSA and PD (around 28–30%).

Nevertheless, the prevalence and incidence of nSH and diagnosed hypertension, as well as their association with nOH, may vary substantially according to age, sex, and the duration of different synucleinopathies. More studies with an in-depth evaluation of these factors are therefore warranted.

## 5. Pathophysiology of Neurogenic Supine Hypertension

From a pathophysiological standpoint, in addition to mechanisms typically implicated in primary hypertension, and the use of drugs that are deployed to counteract nOH such as fludrocortisone and midodrine, nSH in autonomic disorders arises from distinct processes that can diversely affect the afferent, central and efferent Synthetic Nervous System pathway of the arterial baroreflex arch, denervation supersensitivity, caused by altered sympathetic transmission that can impair adrenoceptors, and even the renin–angiotensin–aldosterone system [2,7]. Moreover, between these neurodegenerative disorders some differences in these mechanisms can be detected. In MSA, which is characterized by a central deregulation of autonomic output, residual sympathetic tone in peripheral nerves may explain the etiology of supine hypertension (SH). The cause of SH in pure autonomic failure (PAF) is less clear. However, it has been suggested that increased vascular resistance, persistent sympathetic tone, and heightened sensitivity to norepinephrine at postsynaptic membranes may contribute to SH in these patients [29]. Regarding the renin–angiotensin–aldosterone system, angiotensin II may also play a key role in SH. Elevated levels of angiotensin II have been observed in patients with autonomic failure compared to healthy individuals, despite low renin levels and normal aldosterone concentrations [30,31]. This supports the hypothesis of a renin-independent mechanism contributing to angiotensin II formation [32]. Additionally, reduced nitric oxide (NO) bioavailability or production has been proposed as a pathophysiological mechanism contributing to SH. Conversely, an exaggerated vasodilatory response to NO-releasing drugs, such as sildenafil and nebivolol, has been reported in patients with dysautonomia. This suggests a possible increased sensitivity to nitric oxide in these individuals [33].

Thus, the open question remains: is it worth it to treat nSH, when present, in patients with alpha-synucleinopathies? Even in the presence of nOH? An overview of studies about hypertension-mediated organ damage in these patients may help to give a response.

## 6. Neurodegenerative Synucleinopathies and Organ Damage

### 6.1. Carotid Intima-Media Thickness

Several case–control studies with relatively small sample sizes investigated carotid intima-media thickness (cIMT) in patients with Parkinson’s disease (PD) compared to control groups. Most of these studies reported lower cIMT values in PD patients, suggesting a potential “protective” vascular effect in this population. For example (see Table 1, also), Lee et al. [34] found that the average cIMT in PD patients (*n* = 43; mean age 68 years; 23.3% hypertensive) was more than 0.11 mm lower than in a control group matched not only for age and sex, but also for cardiovascular risk factors. Interestingly, within the PD group, cIMT was inversely associated with both levodopa treatment duration and disease severity, as measured by the Hoehn–Yahr stage. The authors hypothesized that levodopa, known for its antihypertensive effects, may contribute to this observed vascular difference.

Subsequent studies have replicated these findings. For instance, Zambito Marasla et al. confirmed significant differences in cIMT between PD patients and controls, even in older cohorts and when analysing left and right carotid arteries separately [35]. However, contradictory evidence also exists. In a study by Alexa et al., PD patients exhibited increased cIMT compared to controls [38]. Similarly, Alves et al. found no significant difference in cIMT, but reported a higher prevalence of carotid plaques in the PD group—possibly explained by disrupted nocturnal dipping patterns, with 50% of patients showing altered circadian BP profiles on ambulatory blood pressure monitoring (ABPM) [36].

Conversely, Yan et al. found no differences in either cIMT or plaque prevalence between PD patients and controls [37].

Finally, a longitudinal study by Rektor et al. examined the hypothesis that in PD patients, the presence of carotid atherosclerosis and intracerebral vascular damage may be associated with increased mortality, suggesting that vascular burden could influence disease prognosis [39]. In summary, available data do not permit firm conclusions regarding whether cIMT or carotid plaque burden are elevated or reduced in individuals with synucleinopathies.

### 6.2. Arterial Stiffness

In a comparative study involving 70 patients with Parkinson’s disease (PD) and 77 healthy controls, no significant differences were observed in peripheral blood pressure (BP). However, central diastolic BP was significantly higher in the PD group, along with a trend toward elevated augmentation index at 75 bpm (AIx@75), suggesting increased central vascular load [42].

In a larger sample of 125 PD patients, arterial stiffness, as measured by brachial-ankle pulse wave velocity (PWV), was significantly increased only in patients with autonomic failure. These increases were associated with orthostatic hypotension, supine hypertension, nocturnal hypertension, and non-dipping BP patterns, highlighting the role of dysautonomia in vascular changes [43].

Interestingly, in another case–control study comparing patients with type 2 diabetes, hypertension, and Parkinson’s disease (PD), the latter group showed pulse wave velocity (PWV) values similar to those with hypertension but higher than those with type 2 diabetes [41].

Using the cardio-ankle vascular index (CAVI) to assess arterial stiffness, Suzuki et al. found that while CAVI values were higher in diabetic or hypertensive controls compared to controls without cardiovascular (CV) risk factors, no such differences were detected among PD patients, regardless of the presence of traditional CV risk factors [44]. This finding raises the hypothesis that PD-specific mechanisms might attenuate the usual impact of CV risk factors on arterial stiffness [44].

In another study using 24 h ambulatory pulse wave analysis with the Mobil-O-Graph™ (I.E.M.), PWV was significantly elevated and AIx@75 significantly reduced in 32 patients (27 with PD and 5 with MSA), compared to 15 healthy controls. Additionally, central systolic BP and mean nocturnal peripheral systolic BP were significantly elevated in the patient group. Notably, over 80% of these patients presented with either nocturnal hypertension or a non-dipping pattern, suggesting a potential interaction between autonomic dysregulation and arterial stiffness [45].

Arterial stiffness markers have also been evaluated in response to physiological stress. For example, after a 30- or 60-degree head-up tilt test, both AIx and PWV decreased substantially in 10 patients with autonomic failure. These changes were closely correlated with concurrent BP reductions, indicating that these stiffness indices are highly sensitive to real-time hemodynamic changes [62].

Exercise-related changes in arterial stiffness were assessed in 20 PD patients by Kanegusuku et al. [63]. After self-selected intensity aerobic exercise, no change in PWV was observed. In contrast, after a non-exercise control session, brachial and central systolic BP/diastolic BP, AIx, and heart rate all increased, suggesting that moderate-intensity exercise may help stabilize arterial stiffness and hemodynamics in PD patients.

In an integrated comparative study pooling patients with autonomic failure (5 MSA, 7 PAF, 15 PD) and comparing them with patients with essential hypertension and normotensive healthy controls, Milazzo and colleagues found that the left ventricular hypertrophy (LVH) and PWV in the autonomic failure group were comparable to those observed in the essential hypertension group and significantly higher than in healthy controls. However, AIx@75 and central BP were significantly higher in the autonomic failure group than in hypertensive patients, suggesting a distinct vascular phenotype likely driven by autonomic dysfunction [40].

Thus, even for arterial stiffness, different studies have reported divergent results, which may partly reflect the use of different measurement methods. Some authors also propose a prominent role for autonomic failure, in addition to traditional risk factors such as hypertension and diabetes.

### 6.3. Renal Function and Parkinson’s Disease or Other Alpha-Synucleinopathies: Evidence from Observational and Longitudinal Studies

Several case–control and cross-sectional studies have explored the relationship between renal function and diseases characterized by autonomic failure, including PD, Pure Autonomic Failure (PAF), and Multiple System Atrophy (MSA).

In a study comparing 200 PD patients and 110 healthy controls, multivariate logistic regression revealed that blood urea nitrogen, serum creatinine, and urine protein levels were independently associated with PD, suggesting potential renal involvement or systemic metabolic alterations in PD [46].

In a cohort of 64 men with PAF, renal dysfunction—manifested as increased serum creatinine, decreased estimated glomerular filtration rate (eGFR), and elevated blood urea nitrogen—was observed only in those with SH, indicating a possible link between nocturnal BP overload and renal impairment [48].

Similarly, in the previously cited study by Palma et al., involving a mixed group of patients with autonomic failure (including PD, MSA, and PAF), reduced eGFR was significantly associated with the presence of SH, reinforcing the hypothesis that blood pressure dysregulation in autonomic disorders contributes to target organ damage, particularly the kidneys [51].

Beyond observational associations, longitudinal cohort studies have assessed whether impaired renal function may precede or contribute to the risk of developing PD. In a large-scale Korean cohort of nearly 3.5 million individuals aged ≥65 years, followed for a mean of 5.2 years, both reduced eGFR and dipstick-positive proteinuria were independently associated with increased risk of incident PD, after adjusting for confounders. Notably, the hazard ratio (HR) for PD was highest in those with both reduced eGFR and proteinuria, indicating a cumulative effect of kidney dysfunction on neurodegeneration risk [47].

In a subsequent study using the same Korean cohort, but with a different design, Kwon et al. examined 16,000 individuals with clinically diagnosed CKD versus over 66,000 control subjects aged ≥40 years. Their findings suggested that CKD was significantly associated with PD, and this association was influenced by contextual factors such as rural residence, normoweight status, and fasting blood glucose ≥100 mg/dL, implying possible environmental and metabolic mediators [50].

Consistent with these findings, an analysis of 400,571 participants from the UK Biobank, followed for over 13 years, revealed a non-linear relationship between low eGFR (<30 mL/min/1.73 m^2^)—estimated from both serum creatinine and cystatin C—and increased PD incidence. This supports the notion that advanced renal dysfunction may play a contributory role in PD pathogenesis, possibly via shared vascular, inflammatory, or metabolic mechanisms [49].

Thus, collectively these studies show an association between kidney function and alpha-synucleinopathies but especially point out that kidney failure often precedes PD.

### 6.4. Cerebral Lesions (White Matter Hyperintensities)

There is ongoing debate about whether white matter hyperintensities (WMH) should be considered a result of hypertension (or even hypotension)-mediated organ damage, or if they are linked to degenerative processes occurring in oligodendrocytes.

Umoto, Lim, and colleagues, in two similar case–control studies, found that compared to patients with Parkinson’s disease (PD) and healthy controls, patients with multiple system atrophy (MSA) had either greater WMH [54] or a greater mediating effect of WMH [52]. Moreover, between PD patients, those with dementia had larger WMH with respect to those with normal cognition [56]. Interestingly, WMH score was independently associated with both systolic blood pressure (BP) [52,54] and with the drop in orthostatic BP [54]. This latter association was also confirmed in another study involving MSA patients [53]. Other studies in patients with autonomic failure found that either nocturnal hypertension or supine hypertension was associated with WMH [51,55,57]. In a longitudinal study of PD patients followed for four years, the progression of WMH was greater in those who died [39].

Therefore, while there is convincing evidence of a strict relationship between hypertensive states (and possibly also hypotensive states) and WMH, most studies are cross-sectional. As a result, causality cannot be established.

### 6.5. Left Ventricle Hypertrophy

As early as the early 2000s, patients with autonomic failure, including PAF and MSA, were recognized to be at risk for developing left ventricular hypertrophy (LVH) [59]. This observation was later confirmed in patients with PD [61]. In that case–control study, which compared 50 PD patients with 50 healthy controls, not only was LVH more prevalent in the PD group, but also left atrial volume and left ventricular filling pressure were increased. Additionally, concentric remodeling and LVH were correlated with advanced PD stages, as determined by the Hoehn and Yahr scale.

However, using healthy normotensive controls as comparators may not fully capture the pathophysiological context. A study by the Turin group [58] found that the prevalence of LVH was similar between patients with autonomic failure and supine hypertension and those with essential hypertension, suggesting that hypertension—regardless of etiology—is the primary driver of LVH, not the underlying neurologic condition itself.

In PD patients, nocturnal blood pressure (BP) patterns provide further insight. When PD patients are stratified by dipping status, LV mass is elevated only in reverse dippers, and is comparable to that in hypertensive controls [60]. As expected, nocturnal BP levels and nocturnal BP load were the strongest correlates of increased LV mass.

Supporting this, an observational study including 57 patients (35 MSA, 14 PD, and 8 PAF) reported a higher prevalence of LVH in those with supine hypertension [51]. These patients also exhibited lower estimated glomerular filtration rate (eGFR) and greater white matter hyperintensity (WMH) volume, both markers of hypertensive organ damage (HMOD). Importantly, during a nearly 2-year follow-up, supine hypertension was independently associated with an increased incidence of cardiovascular events and mortality, underscoring its clinical relevance [51].

Indeed, when considering LVH, haemodynamic factors may exert a stronger influence than the underlying synucleinopathy.

### 6.6. Critical Summary About Hypertension-Mediated Organ Damage in Neurodegenerative Synucleinopathies

The studies about HMOD collectively emphasize that it is not PD or autonomic failure syndromes per se that directly lead to target organ damage. Rather, it is the presence of hypertension, particularly supine or nocturnal hypertension, that constitutes the key factor. As with essential or other forms of hypertension, the presence of HMOD serves as a red flag for heightened cardiovascular risk and necessitates careful evaluation and management [5].

## 7. Clinical Implications

The coexistence of nSH and nOH poses significant therapeutic challenges. Most antihypertensive agents risk exacerbating OH, thereby limiting their use. Consequently, physicians often rely on non-pharmacologic interventions (e.g., head-of-bed elevation, salt restriction timing, and compression garments) that offer some benefit with minimal risk.

Given these complexities, treatment efforts traditionally focus on managing nOH, often overlooking nSH—despite evidence suggesting that supine hypertension may worsen nOH via augmented nocturnal natriuresis, thereby perpetuating the cycle of volume depletion and OH [7].

## 8. Modification of Life Style

The implementation of antihypertensive therapy is often limited due to its potential to exacerbate nOH, making it difficult to establish a clear treatment strategy.

Behavioral measures are more easily recommended and typically well tolerated, but there is often a low adherence and yet they rarely succeed in reducing severe supine or nocturnal hypertension to acceptable levels. Among non-pharmacological strategies, particular attention should be given to addressing sleep-related disturbances, such as obstructive sleep apnea syndrome (OSAS) and fragmented sleep, which are known to aggravate nocturnal or supine hypertension [2,3].

A practical intervention includes sleeping with the bed tilted at a 12° head-up angle, or even with the entire bed tilted, which imposes a mild gravitational stress. This posture can reduce nocturnal BP and mitigate nocturnal diuresis, which contributes to volume depletion and worsens nOH [2,3]. Similarly, avoiding the supine position during the day can help limit the amount of time spent with elevated blood pressure. Very recently, two proof-of-concept studies have investigated whether other non-pharmacological interventions—such as overnight application of continuous positive airway pressure or localized heat (40–42 °C applied with a heating pad over the abdomen)—can reduce blood pressure in patients with autonomic failure and nSH. These studies have shown encouraging results [64,65]. Interestingly, the postprandial drop in BP, typically considered a concern, may actually be harnessed therapeutically in these patients to reduce nSH when meals are scheduled appropriately [66].

Conversely, to mitigate orthostatic hypotension, behavioral interventions such as adequate hydration, increased salt intake, and the use of compression garments (e.g., abdominal binders or stockings) can be effective [2,3], even if in the case of salt intake, the recommendation is grounded in physiological mechanisms rather than supported by clinical trials specifically involving patients with synucleinopathies. However, it is crucial to regularly review pharmacological treatments, as medications commonly used to treat OH—such as midodrine and fludrocortisone—can worsen supine hypertension and complicate the overall management plan [2,3].

## 9. Antihypertensive Therapy

Gibbons and an expert panel suggested a stepwise approach to the diagnosis of nOH, using home BP and BP measurement during tilting, but it is less clear how to approach nSH. Higher cut-offs, like 160–180 mmHg of systolic BP are proposed and the use of Ambulatory Blood Pressure monitoring rather than simple random or home-based BP measurements is recommended [3]. Moreover, the immediate danger of a worsening of nOH is weighed against the possible long-term effect of high BP.

The consensus by the American Autonomic Society (AAS) and the European Federation of Autonomic Societies (EFAS), focuses more on nSH, and suggests a careful use of home blood pressure measurements in different body positions and different timing, other than repeated ABPM. nSH is defined as systolic BP ≥ 140 mmHg and/or diastolic BP ≥ 90 mmHg, measured after at least 5 min of rest in the supine position.

In both consensus documents, no mention of HMOD is made. If behavioural changes or the modulation of midodrine or fludrocrtisone are not sufficient to improve BP control, the use of short-acting agents such as captopril, losartan or clonidine is advocated but based only on pathophysiology and not on scientific evidence.

On the contrary, hypertension-specific guidelines from scientific societies such as the ESH, ESC, and AHA/ACC place significant emphasis on the assessment of HMOD [5,67,68]. This approach, incorporated into the classification of hypertension by grades and stages, helps better discriminate which patients are at high or very high CV risk. Although these guidelines are not specifically tailored to patients with neurodegenerative disorders, they apply to all hypertensive patients, including those with such conditions. Therefore, they can provide valuable guidance in making the challenging decision of whether or not to initiate antihypertensive treatment. As always, the decision should be individualized and discussed with the patient, just as it would be for any hypertensive patient.

Another unresolved issue concerns the optimal blood pressure target that should be pursued once treatment is initiated in patients with nSH. It is important to emphasize that individuals with neurodegenerative synucleinopathies are generally frail and at high risk of falls. Therefore, overly aggressive blood pressure targets should be avoided, and treatment goals should be individualized [66].

## 10. Conclusions

In conclusion, although studies on HMOD in patients with these neurodegenerative disorders are limited in size and design, they suggest that HMOD can provide valuable information. However, it remains uncertain whether this information can reliably guide decisions regarding the initiation of antihypertensive treatment or determine appropriate tolerance thresholds. Nevertheless, as for simple hypertensive patients, the presence of HMOD in a patient with hypertension and a synucleinopathy is likely associated with a worse prognosis. Experts agree that the decision to start antihypertensive therapy should be individualized, taking into account the patient’s neurological prognosis and the anticipated impact on neurogenic orthostatic hypotension (nOH). We believe that—as with other hypertensive patients—careful assessment of HMOD can yield meaningful insights and should be performed systematically in dysautonomic patients with nSH. The use of short half-life medications at bedtime appears reasonable, although there is still debate over which drug class should be preferred.

## Figures and Tables

**Table 1 biomedicines-14-00040-t001:** Most relevant studies featuring HMOD in patients with neurodegenerative synucleinopathies.

Study	HMOD	Sample Size and Type of Patients	Design	Conclusion
Lee [34]	cIMT & carotid plaques	43 PD vs. 86 matched HC	case–control	cIMT was lower in PD than in HC
Zambito Masala [35]	cIMT & carotid plaques	30 PD vs. 30 HC	case–control	cIMT was lower in PD than in HC
Alves [36]	cIMT & carotid plaques	102 PD vs. 102 HC	case–control	No difference in cIMT but more plaques in PD
Yan [37]	cIMT & carotid plaques	68 PD vs. 81 HC	case–control	No difference in cIMT and plaques between PD and HC
Alexa [38]	cIMT & carotid plaques	54 PD vs. 50 HC	case–control	cIMT was higher in PD than in HC
Rektor [39]	cIMT & carotid plaques	57 PD whose 18 died	longitudinal	More-severe vascular impairment in deceased patients
Milazzo [40]	arterial stiffness	27 AF (5 MSA + 7 PAF + 15 PD) vs. 27 HT + 27 HC	case–control	cf-PWV lower in HC than both in MSA+PAF and EH
Awassi Yuphiwa [41]	arterial stiffness	50 (PD vs. HT vs. DMT2)	case–control	T2DM has lower PWV than PD
Park [42]	arterial stiffness	70 PD 77 HC	case–control	No significant difference in the AIx between PD and HC
Kim [43]	arterial stiffness	125 PD vs. 22 HC	case–control	ba-PWV higher only in PD with autonomic failure
Suzuki [44]	arterial stiffness	63 PD vs. 63 HC	case–control	ba-PWV was smaller in PD than HC
Franzen [45]	arterial stiffness	32 PTS (27 PD, 5 MSA) 15 HC matched for CVRF	case–control	Aix@75 was lower and PWV higher in PD+MSA+PAF vs. HC
Yang [46]	renal function	200 PD vs. 110 HC	case–control	Elevated creatinine, and u-protein associates with PD
Nam [47]	renal function	3,580,435 individuals aged ≥65 years who had undergone health checkups	longitudinal	eGFR associates with increased risk of PD
Garland [48]	renal function	64 male pts with PAF whose 48% with supine HT (Systolic BP > 150 mmHg)	case–control.	eGFR lower in pts with nSH
Peng [49]	renal function	400,571 UK Biobank participants	long.	eGFR associates with increased risk of PD
Kwon [50]	renal function	Korean National Health Insurance Service-Health Screening Cohort (16,559 with CKD & 66,236 control)	longitudinal	The general CKD population may not exhibit a greater propensity for PD than their non-CKD counterparts
Palma [51]	renal function	57 (35 MSA, 14 PD & 8 PAF)	observational	PD+MSA+PAF with nSH has lower eGFR than PD+MSA+PAF without nSH
Lim [52]	WMH	63 MSA vs. 63 age- and sex-matched HC	case–control	The median grading score of WMH was higher in MSA than in HC
Tha [53]	WMH	16 MSA vs. 16 HC	case–control	WMH in patients with MSA were significantly larger that in HC
Umoto [54]	WMH	22 MSA vs. 22 PD vs. 22 HC	case–control	WMHs in patients with MSA were statistically greater than in PD patients or HC
Oh [55]	WMH	129 PD patients (27 normal BP dipping; 102 non-dipping)	observational	Nocturnal hypertension (not non-dipping) associates with increased WMH score
Palma [51]	WMH	57 (35 MSA, 14 PD & 8 PAF)	observational	PD+MSA+PAF with nSH has higher WMH volumes than PD+MSA+PAF without nSH
Oh [56]	WMH	53 PD-normal cognition, 76 had PD-mild cognitive impairment), 43 had PD-dementia and 14 had DLB	cross-sectional	U-protein/creatinine ratio and WMH higher in PD with dementia & DLB than in PD with mild cognitive impairment or normal cognition
Chen [57]	WMH	52 PD patients (13 normal BP dipping; 39 non-dipping)		Patients in the non-dipping group exhibited a significantly higher volume of periventricular WMH
Maule [58]	LVH	25 AF (21 MSA, 3 PAF, 1 amylpoidosis) vs. 20 HT	case–control	LVM is similar in patients with autonomic failure and HC
Vagaonescu [59]	LVH	6 PAF & 8 MSA vs. 14 HC	case–control	LVH lower in HC than both in MSA+PAF and EH
Di Stefano [60]	LVH	26 PD with reverse dipping + 26 PD without reverse dipping + 26 HT	case–control	LVH higher in PD with reverse BP dipping than PF without reverse dipping
Piqueras-Flores [61]	LVH	50 PD vs. 50 HC	case–control	LVMi higher in PD than HC
Palma [51]	LVH	57 (35 MSA, 14 PD & 8 PAF)	observational	PD+MSA+PAF with nSH has higher LVM than PD+MSA+PAF without nSH
Milazzo [40]	LVH	27 AF (5 MSA+7 PAF + 15 PD) vs. 27 HT + 27 HC	case–control	LVH lower in HC than both in AF and EH

## Data Availability

No new data were created or analyzed in this study. Data sharing is not applicable to this article.

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
