# Peer review of "Should Neurogenic Supine Hypertension Be Treated? Insights from Hypertension-Mediated Organ Damage Studies—A Narrative Review"

_biomedicines, 2025, doi:10.3390/biomedicines14010040_

Round 1

Reviewer 1 Report

Comments and Suggestions for Authors

This is an excellent review highlighting the clinical challenges of managing complex hypertension in patients with neurological diseases, with a particularly valuable focus on the often overlooked problem of neurogenic supine hypertension. I have several comments and suggestions:

  1. Context of intensive BP control trials and guidelines. Given the increasingly aggressive BP treatment goals recommended based on major trials such as SPRINT, it would be useful to discuss the implications of these targets for managing neurogenic supine hypertension in a broader clinical context.
  2. Epidemiology of autonomic failure and neurogenic supine hypertension. In the section on the “epidemiology of autonomic failure disorders,” the presented data are somewhat broad and focus mainly on the prevalence of neurodegenerative diseases. If available, it would strengthen the review to include data on: The overlap between patients with neurogenic supine hypertension and those with diagnosed hypertension; The proportion of patients who have both nOH and neurogenic supine hypertension; and How these overlaps vary by age group and which subgroups (or clinical characteristics) are at highest risk. If such data are not available, it may be worthwhile to explicitly highlight this as an important knowledge gap and a priority for future research.
  3. Hypertension-mediated target organ damage and synthesis of mixed findings. The authors review the role of hypertension-mediated target organ damage—such as carotid intima–media thickness, arterial stiffness, and renal dysfunction—in several sections. In each, the evidence cited often shows mixed or inconsistent findings across studies. A brief concluding synthesis at the end of each relevant section, explicitly interpreting and reconciling the disparate results, would help readers better understand the overall weight of the evidence and the key areas of uncertainty.
  4. The authors noted “Interestingly, the postprandial drop in BP, typically considered a concern, may actually be harnessed therapeutically in these patients to reduce nSH when meals are scheduled appropriately.” Are there scientific data on this recommendation? It would be helpful to provide the citations or be clear this is still a speculation that needs to be tested.
  5. Similarly, the authors also mentioned increasing salt intake for managing OH. Are there studies that rigorously test this?

Author Response

This is an excellent review highlighting the clinical challenges of managing complex hypertension in patients with neurological diseases, with a particularly valuable focus on the often overlooked problem of neurogenic supine hypertension. I have several comments and suggestions:

We thank the reviewer for the thorough evaluation of our manuscript, the positive comments, and the important suggestions provided.

  1. Context of intensive BP control trials and guidelines. Given the increasingly aggressive BP treatment goals recommended based on major trials such as SPRINT, it would be useful to discuss the implications of these targets for managing neurogenic supine hypertension in a broader clinical context.

We agree with the reviewer that this point should be emphasized. We have added a comment regarding the BP target (please see page 9, lines 428-432).

  1. Epidemiology of autonomic failure and neurogenic supine hypertension. In the section on the “epidemiology of autonomic failure disorders,” the presented data are somewhat broad and focus mainly on the prevalence of neurodegenerative diseases. If available, it would strengthen the review to include data on:

The overlap between patients with neurogenic supine hypertension and those with diagnosed hypertension; The proportion of patients who have both nOH and neurogenic supine hypertension; and How these overlaps vary by age group and which subgroups (or clinical characteristics) are at highest risk. If such data are not available, it may be worthwhile to explicitly highlight this as an important knowledge gap and a priority for future research.

Actually, we summarized the available information on this topic in the section entitled ‘Parkinson’s Disease or other α-synucleinopathies and hypertension incidence/prevalence’, but we also acknowledged that more specific data in relevant subgroups are warranted (please see pages 3-4, lines 150-154).

  1. Hypertension-mediated target organ damage and synthesis of mixed findings. The authors review the role of hypertension-mediated target organ damage—such as carotid intima–media thickness, arterial stiffness, and renal dysfunction—in several sections. In each, the evidence cited often shows mixed or inconsistent findings across studies. A brief concluding synthesis at the end of each relevant section, explicitly interpreting and reconciling the disparate results, would help readers better understand the overall weight of the evidence and the key areas of uncertainty.

Thank you for the suggestion. We added a short summary to each HMOD section and included a separate section to highlight the overall HMOD considerations (please see page 5, lines 209-211; page 6, lines 259-262; page 7 lines 349-350). Please note that a comments were already present for renal function and WMH (please see page 6, lines 301-303; page 7, lines 322-324).

  1. The authors noted “Interestingly, the postprandial drop in BP, typically considered a concern, may actually be harnessed therapeutically in these patients to reduce nSH when meals are scheduled appropriately.” Are there scientific data on this recommendation? It would be helpful to provide the citations or be clear this is still a speculation that needs to be tested.

There are several studies that have described and attempted to explain this phenomenon both in healthy individuals and in patients with dysautonomia. We have added a recent study (see new reference n. 66) that focuses specifically on patients with PD and MSA.”

  1. Similarly, the authors also mentioned increasing salt intake for managing OH. Are there studies that rigorously test this?

There are numerous studies, reviews, and recommendations that can be cited to support this statement. However, we must acknowledge that rigorous trials specifically testing whether increasing salt intake is beneficial for nOH in patients with synucleinopathies are lacking. We have emphasized this point in the manuscript as well (see the passage on page 8  lines 395-397)

Reviewer 2 Report

Comments and Suggestions for Authors

This is a comprehensive, critical and extensive review of studies documenting hypertension-mediated end-organ damage (HMOD) associated with the presence of supine hypertension (nSH) in patients with neurogenic orthostatic hypotension due to synucleinopathies.

Simply putting together and summarizing this information is a valuable contribution. The authors argue that knowing if a given patients has HMOD can assist in deciding how much effort providers should dedicate in managing nSH.

I have the following comments:

  1. The section on organ damage (lines 172-330) provides a comprehensive review of numerous studies which is very valuable. I fear, however, that the reader may be confused by the recitation of so many studies. It would be helpful, therefore, if the authors provide a critical summary of the findings. I assume that this is done on lines 325-330, but a titled section would be helpful.
  2. Line 391. I assume the authors mean to say that the presence of HMOD can be associated with worse prognosis. The way the sentence is phrased, however, may be interpreted as the opposite: …”there is no reason to think that…the presence of HMOD…can be associated with a worst prognosis.” Please clarify.
  3. Even though the focus of this manuscript is to review HMOD rather than the treatment of nSH, the authors may consider mentioning that nSH can be managed by simply avoiding the supine posture during the day. This does not solve the problem of patients that have daytime seated HTN.
  4. There are also “single dose” small studies showing that nonpharmacological treatment of nocturnal hypertension with a heated mattress or CPAP is as effective as bedtime antihypertensives in controlling nocturnal SH and have the added benefit of reducing nocturia. This may provide hope that therapies for nSH can be developed that can allow us to “have our cake and eat it too”.

END

Author Response

This is a comprehensive, critical and extensive review of studies documenting hypertension-mediated end-organ damage (HMOD) associated with the presence of supine hypertension (nSH) in patients with neurogenic orthostatic hypotension due to synucleinopathies.

Simply putting together and summarizing this information is a valuable contribution. The authors argue that knowing if a given patients has HMOD can assist in deciding how much effort providers should dedicate in managing nSH.

We thank the reviewer for the thorough evaluation of our manuscript, the positive comments, and the important suggestions provided. 

I have the following comments:

  1. The section on organ damage (lines 172-330) provides a comprehensive review of numerous studies which is very valuable. I fear, however, that the reader may be confused by the recitation of so many studies. It would be helpful, therefore, if the authors provide a critical summary of the findings. I assume that this is done on lines 325-330, but a titled section would be helpful.

Thank you for the suggestion. In response, we have added a heading to the sentences indicated by the reviewer. (please see page 7 lines 352-353 and following)

  1. Line 391. I assume the authors mean to say that the presence of HMOD can be associated with worse prognosis. The way the sentence is phrased, however, may be interpreted as the opposite: …”there is no reason to think that…the presence of HMOD…can be associated with a worst prognosis.” Please clarify.

           Thank you. We have rephrased the sentence as suggested. (please see page 9, lines 438-440)

  1. Even though the focus of this manuscript is to review HMOD rather than the treatment of nSH, the authors may consider mentioning that nSH can be managed by simply avoiding the supine posture during the day. This does not solve the problem of patients that have daytime seated HTN.

Thank you for the suggestion. We have added a sentence addressing this point. (please see page 8, lines 384-385)

  1. There are also “single dose” small studies showing that nonpharmacological treatment of nocturnal hypertension with a heated mattress or CPAP is as effective as bedtime antihypertensives in controlling nocturnal SH and have the added benefit of reducing nocturia. This may provide hope that therapies for nSH can be developed that can allow us to “have our cake and eat it too”.

We thank the reviewers for these suggestions. We agree that, although preliminary, the results of these studies are nevertheless interesting and worth sharing (please see page 8, lines 385-390)

Reviewer 3 Report

Comments and Suggestions for Authors

Comments to the authors:
In the manuscript “Should Neurogenic Supine Hypertension Be Treated? Insights from Hypertension-Mediated Organ Damage studies- A narrative review” the authors provided a detailed review of the literature and current knowledge regarding synucleinopathies – a group of neurodegenerative disorders characterized by neuronal or glial inclusions, composed of abnormal accumulation of α-synuclein protein – that are often linked to cardiovascular autonomic dysfunction. This commonly leads to neurogenic orthostatic hypotension, which may occur alone or in combination with neurogenic supine hypertension.
The authors discussed various trial data and guidelines and highlighted that the treatment decisions should be individualized. The authors concluded that antihypertensive therapy may help prevent subclinical organ damage, but clinicians must balance this benefit against the risk of worsening neurogenic orthostatic hypotension and further compromising overall prognosis.
The manuscript is comprehensive, clearly written with logically organized sections, and overall easy to follow. However, subtitles need to be improved:
In line 51: remove abbreviation “(SH)” from the subtitle “Definition of supine hypertension (SH) in the context of autonomic failure”.
In line 70: “EPIDEMIOLOGY OF AUTONMIC FAILURE DISORDERS” rewrite in lowercase, and correct spelling in “autonomic”
In line 103: “PD or other alpha-synucleinopathies and hypertension incidence/prevalence” give full name for abbreviation “PD”.
In line 146: in subtitle “Pathophysiology” please incorporate “Pathophysiology of what condition/disease”.
In line 172: “PD and organ damage” give full name instead of abbreviation for “PD”.
In line 173: give full name for abbreviation “cIMT”.
In line 285: “Cerebral Lesions (WMH)” please give full name for abbreviation WMH in brackets.
In line 301: ”LVM” please give full name instead of this abbreviation.
The authors reasonably discussed data from various observational and longitudinal studies and novel clinical trials.
The authors draw appropriate and reasonable conclusions from collected scientific data in this review manuscript.
The authors presented relevant studies, data and most important conclusions from a study in an easy-to-understand manner in Table 1.
References cited in the text are appropriate and adequate, 62% cited references were published in recent 10 years, with 23 references published in last 5 years. However, a reference no. 63 “Oh, Y.-S.; Kim, J.-S.; Park, H.-E.; Song, I.-U.; Park, J.-W.; Yang, D.-W.; Son, B.-C.; Lee, S.-H.; Lee, K.-S. Association between Urine Protein/Creatinine Ratio and Cognitive Dysfunction in Lewy Body Disorders. J Neurol Sci 2016, 362, 258–262, 568. doi:10.1016/j.jns.2016.01.061” is only cited in Table 1. The authors should cite and discuss the reference in the text, also.  

Author Response

Comments to the authors:
In the manuscript “Should Neurogenic Supine Hypertension Be Treated? Insights from Hypertension-Mediated Organ Damage studies- A narrative review” the authors provided a detailed review of the literature and current knowledge regarding synucleinopathies – a group of neurodegenerative disorders characterized by neuronal or glial inclusions, composed of abnormal accumulation of α-synuclein protein – that are often linked to cardiovascular autonomic dysfunction. This commonly leads to neurogenic orthostatic hypotension, which may occur alone or in combination with neurogenic supine hypertension.
The authors discussed various trial data and guidelines and highlighted that the treatment decisions should be individualized. The authors concluded that antihypertensive therapy may help prevent subclinical organ damage, but clinicians must balance this benefit against the risk of worsening neurogenic orthostatic hypotension and further compromising overall prognosis.
The manuscript is comprehensive, clearly written with logically organized sections, and overall easy to follow. However, subtitles need to be improved:

We thank the reviewer for the thorough evaluation of our manuscript, the positive comments, and the important suggestions provided.

In line 51: remove abbreviation “(SH)” from the subtitle “Definition of supine hypertension (SH) in the context of autonomic failure”.

It was removed
In line 70: “EPIDEMIOLOGY OF AUTONMIC FAILURE DISORDERS” rewrite in lowercase, and correct spelling in “autonomic”

It was done.
In line 103: “PD or other alpha-synucleinopathies and hypertension incidence/prevalence” give full name for abbreviation “PD”.

It was done.
In line 146: in subtitle “Pathophysiology” please incorporate “Pathophysiology of what condition/disease”.

It was done.
In line 172: “PD and organ damage” give full name instead of abbreviation for “PD”.

It was done and put “Neurodegenerative synucleinopathies” instead.

In line 173: give full name for abbreviation “cIMT”.

It was done.
In line 285: “Cerebral Lesions (WMH)” please give full name for abbreviation WMH in brackets.

It was done.
In line 301: ”LVM” please give full name instead of this abbreviation.

It was done.
The authors reasonably discussed data from various observational and longitudinal studies and novel clinical trials.
The authors draw appropriate and reasonable conclusions from collected scientific data in this review manuscript.
The authors presented relevant studies, data and most important conclusions from a study in an easy-to-understand manner in Table 1.

References cited in the text are appropriate and adequate, 62% cited references were published in recent 10 years, with 23 references published in last 5 years. However, a reference no. 63 “Oh, Y.-S.; Kim, J.-S.; Park, H.-E.; Song, I.-U.; Park, J.-W.; Yang, D.-W.; Son, B.-C.; Lee, S.-H.; Lee, K.-S. Association between Urine Protein/Creatinine Ratio and Cognitive Dysfunction in Lewy Body Disorders. J Neurol Sci 2016, 362, 258–262, 568. doi:10.1016/j.jns.2016.01.061” is only cited in Table 1. The authors should cite and discuss the reference in the text, also.  

The article by Oh and colleagues (now reference n. 56) has also been discussed in the text (please see page 7, lines 314-315)

Round 2

Reviewer 1 Report

Comments and Suggestions for Authors

The authors have fully addressed the reviewer's questions and comments. Thank you!